# Parenting Practices as a Mediator in the Association Between Family Socio-Economic Status and Screen-Time in Primary Schoolchildren: A Feel4Diabetes Study

**DOI:** 10.3390/ijerph15112553

**Published:** 2018-11-14

**Authors:** Sara De Lepeleere, Ilse De Bourdeaudhuij, Vicky Van Stappen, Nele Huys, Julie Latomme, Odysseas Androutsos, Yannis Manios, Greet Cardon, Maïté Verloigne

**Affiliations:** 1Department of Movement and Sports Sciences, Ghent University, Watersportlaan 2, 9000 Ghent, Belgium; sara_de_lepeleere@hotmail.com (S.D.L.); ilse.debourdeaudhuij@ugent.be (I.D.B.); vivstapp.vanstappen@ugent.be (V.V.S.); nele.huys@ugent.be (N.H.); julie.latomme@ugent.be (J.L.); greet.cardon@ugent.be (G.C.); 2Research Foundation Flanders (FWO), Egmontstraat 1, 1000 Brussel, Belgium; 3Department of Nutrition and Dietetics, School of Health Science & Education, Harokopio University, 70 El. Venizelou, 17671 Kallithea, Athens, Greece; oandrou@hua.gr (O.A.); manios@hua.gr (Y.M.)

**Keywords:** TV-time, gaming, parental education, parenting skills

## Abstract

The aim of this study was to examine the mediating effects of specific parenting practices on the association between family socio-economic status (SES) and screen-time of 6- to 9-year-old children from families with an increased risk of developing type 2 diabetes. This cross-sectional study, focusing on families with an increased risk of developing type 2 diabetes, used the Belgian baseline data of the Movie Models intervention, integrated within the European Feel4Diabetes intervention, and included 247 parents (57.6% lower SES family; 78.0% mothers) who completed a questionnaire. Mediating effects were tested using MacKinnon’s product-of-coefficients test via multilevel linear regression analyses. Being consistent concerning rules about gaming (β = 0.127; standard error = 0.055; 95% CI = 0.020; 0.234) and avoiding negative role modeling concerning TV-time (β = −0.082; standard error = 0.040; 95% CI = −0.161; −0.003) significantly mediated the inverse association between family SES and children’s screen-time. Parents from lower SES families were more consistent concerning rules about gaming and watched more TV nearby their child compared to parents from higher SES families, and these parenting practices were related to more screen-time. No other parenting practices were found to mediate this association. Thus, parents from lower SES families with a higher risk for developing type 2 diabetes might limit their own TV-time nearby their child to reduce their child’s screen-time. Future research should examine other possible mediating factors to develop effective interventions targeting this important at-risk group.

## 1. Introduction

The amount of screen-time in children plays an important role in the development of overweight and obesity [1,2] and their related health consequences such as cardiovascular diseases, type 2 diabetes, cancer, depression, and stress [3,4]. However, Belgian 10- to 12-year-old boys and girls spend on average 205 min/day and 178 min/day, respectively, on screen-time (TV and computer activities combined) [5], although current guidelines recommend less than two hours per day of recreational screen-time [6]. In addition, previous research has shown that children from low socio-economic status (SES) families have higher levels of screen-time [2,5,7,8,9,10], suggesting that these children are potentially at an even higher risk of experiencing negative health consequences [11,12,13]. Therefore, it is important to understand the mechanism through which a lower family SES influences children’s screen-time to develop effective interventions targeting this at-risk group.

Parents play an important role in influencing children’s screen-time by adopting specific parenting practices [14]. Specific parenting practices aim to positively influence the child’s behavior, whereas more general parenting styles refer to the broader emotional and relational climate in which these practices occur [14]. There are many parenting practices related to children’s screen-time that can differ for TV-time or gaming [15]. In a recent review, parenting practices and their associations with screen-time in young children (<6 years old) were investigated [15]. Moderate to strong evidence was found for a positive association between parents’ own TV-time and children’s screen-time. In addition, watching TV with their child and having meals when the TV is on were associated with increased screen-time in children. Another study found that 6- to 12-year-old children had a lower screen-time if parents limited their own gaming [16]. 

Furthermore, parenting practices tend to differ according to SES [14]. Mothers from a low SES family are often faced with a wide range of physical and psychosocial stressors that may interfere with their ability to be responsive to their children’s needs. They tend to have lower levels of social support and are more prone to mental health problems that are known to compromise supportive parenting practices [17,18], such as parental rules [19] and parental modeling [20]. The systematic review of Gebremariam et al. investigated the association between socio-economic position and correlates of sedentary behavior among youth between 0 and 18 years old. Socio-economic position was inversely related to the presence of a TV in the child’s bedroom, parental modeling for TV viewing, parental co-viewing, and eating meals in front of the TV. They found no or indeterminate evidence for an association between socio-economic position and rules and regulations about screen-time [10]. 

Thus, parenting practices might mediate the relationship between family SES and children’s screen-time. However, few mediation studies have been conducted so far [10]. A study conducted in 6- to 11-year-old children showed that the time spent in front of a screen together with parents mediated the inverse relation between family SES and children’s screen-time [21]. Another study investigated whether ‘family television environment’ mediated the relationship between maternal education and 6- to 11-year-old children’s TV-viewing. The authors of that study showed that the parenting practices ‘watching TV together with the child’ and ‘restricting how much time the child spends watching TV’ mediated the inverse relation between maternal education and children’s TV-time [22]. However, the number of mediation studies is scarce and they have investigated only a few factors as potential mediators, indicating that more research is needed. Only by gathering sufficient evidence on the underlying mechanism can effective interventions be developed for low SES families.

Therefore, the aim of this study was to investigate the mediating effect of specific parenting practices on the relation between parental educational level (which is often used as a proxy for family SES) and young primary schoolchildren’s (aged 6 to 9 years old) screen-time. We focused specifically on a population with children from vulnerable families (i.e., families with an increased risk of developing type 2 diabetes), as this study was part of the Feel4Diabetes study (which is further explained in the Materials and Methods section below).

## 2. Materials and Methods

### 2.1. Study Design and Setting

This cross-sectional study used the Belgian baseline data of the Movie Models intervention which was integrated in the European Feel4Diabetes (F4D) intervention. The F4D intervention is a school- and community-based, family-involving intervention aiming to promote a healthy lifestyle and tackle obesity and obesity-related metabolic risk factors for the prevention of type 2 diabetes in families from low income countries (Bulgaria and Hungary), and families living in low socio-economic areas from high income countries (Belgium and Finland) as well as countries under austerity measures (Greece and Spain) in Europe. The F4D intervention was implemented for a two-year period (2016–2018) and included three different components (i.e., a “high-risk families” component, school component, and community component). The “high-risk families” component was only developed for high-risk families, which are defined as families in which at least one parent was identified to be at risk for developing type 2 diabetes (predicted by the FINDRISC score or Finnish Diabetes Risk Score, which is based on age; body mass index; waist circumference; physical activity; dietary consumption of fruits, vegetables, and berries; use of antihypertensive medications; history of high blood glucose; and family history of diabetes) [23]. In the first intervention year in Belgium, these high-risk parents received six counseling sessions (November 2016–March 2017) in which the Movie Models intervention [24] was integrated. The Movie Models intervention targets specific parenting practices related to children’s healthy diet, physical activity, and sedentary behavior, and thus promotes healthy behavior in families. More details on the Movie Models and F4D interventions can be found elsewhere [24,25]. 

### 2.2. Participants and Recruitment

As we used the baseline data from the Movie Models intervention that was integrated within the F4D intervention, we here describe the recruitment procedure of the larger experimental study. However, it should be noted that parents from both the intervention and control groups were included in our study sample without making any distinction between groups. A standardized, multi-stage sampling approach was applied for the recruitment of families from the provinces East- and West-Flanders (Belgium). For each municipality located within a radius of 40 km from Ghent (i.e., the location of the university), the SES was calculated based on the percentage of unemployed people living in the municipality (low/medium/high). Only the municipalities with a low SES (range of unemployed people = 5.5–9.1%) were selected and, based on their number of schools and geographical position, randomly allocated to either the intervention (three municipalities) or the control group (eight municipalities). Next, all schools within the selected municipalities were contacted by telephone and invited to participate (46 intervention schools; 47 control schools). In February 2016, parents of all primary schoolchildren of the first, second, and third grades (6–9 years old) from the schools that agreed to participate (33 intervention schools; 31 control schools) received a questionnaire on socio-demographic characteristics, children’s health behavior, and the FINDRISC questionnaire to determine their risk of developing type 2 diabetes (response rate = 33.5% = 1797/5367 parents, of which 458 were high-risk). In November–December 2016, the ‘high-risk families’ from both the intervention and control groups filled out a questionnaire on parenting practices for the baseline measure of the Movie Models intervention (response rate = 53.9% = 247/458). In total, 247 parents completed both questionnaires and could therefore be included in the analyses.

### 2.3. Measures

#### 2.3.1. Demographic Variables

Socio-demographic variables included children’s and parents’ age and gender, and parental educational level (years of studies). A lower family SES was determined as both parents or one parent having no higher education (less than 14 years) and a higher family SES as both parents having higher education (14 years or more) [26]. 

#### 2.3.2. Children’s Screen-Time 

Screen-time of the child was measured using the question ‘How many hours per day does your child spend on screen-time (screen-time at school not included)?’. It was specified in the questionnaire that screen-time includes all activities that are related to watching TV and DVDs; using the computer, smartphone, and tablet; and playing videogames. Parents were asked to report this separately for weekdays and weekend days. By summing the weekday hours (multiplied by five) and the weekend day hours (multiplied by two) and dividing the sum by seven, the variable ‘daily hours spent on screen-time’ was calculated.

#### 2.3.3. Specific Parenting Practices 

The questionnaire on parenting practices contained a broad range of parenting practices, derived from previously validated questionnaires including the Parental Support for Physical Activity Scale (test-retest reliability: ICC = 0.81) [27], the Parenting Strategies for Eating and Activity Scale [28], and the Parental Feeding Style Questionnaire (test-retest reliability: ICC = 0.76–0.83) [29]. In our questionnaire, 12 parenting practices were specifically related to screen-time (assessed via single items) and were therefore used in the analyses. All items were assessed on a five-point Likert scale: (1) Never, (2) Mostly Not, (3) Sometimes/Sometimes Not, (4) Mostly, (5) Always. For some questions, ‘Not Applicable’ was an alternative answer category, for which the results were set as missing values. For all variables, a higher mean value represents a higher form of the variable. An overview of the items can be found in Table 1. 

### 2.4. Data Analysis

Linear regression analyses were performed using SPSS version 22.0 (SPSS version 22.0, IBM corp., Armonk, NY, USA; 2011). Clustering at the school and municipality level was taken into account by conducting multilevel analyses. Normal distribution of variables was checked by testing skewness and kurtosis. After square transformation of the variable ‘Modeling concerning gaming’, a normal distribution was obtained. Because of their potential relationship with children’s screen-time, children’s age and gender were included as covariates in all analyses and 95% confidence intervals (CI) were reported. 

The mediation analyses consisted of the following steps. Firstly, the main association or direct association between family SES and children’s screen-time was examined (τ-coefficient or c-path). In the second stage, the mediating role of specific parenting practices related to screen-time was examined using the product-of-coefficients test of MacKinnon et al. [30]. This test included the following steps: (1) estimation of the associations between family SES and potential mediators (Action Theory test; α-coefficients or a-path); (2) estimation of the associations between the potential mediators and children’s screen-time (Conceptual Theory Test; β-coefficients or b-path), adjusting for family SES; and (3) calculation of the product of-coefficients (αβ) or c’-path, representing the mediated effect. The rationale behind this third step is that the mediation depends on the extent to which the predictor is related to the mediators, and the extent to which the mediators affect the outcome. After conducting these three steps, statistical significance of the mediated effect was estimated by dividing αβ by its standard error (SE), from which the outcome conforms to a z-distribution. If the ratio was more than 1.96 or smaller than −1.96, the indirect effect was indeed significant, which implied mediation. To calculate SE, the Sobel test was used: SE (αβ) = √(α^2^ *SE (β)^2^ + ^β2^ *SE (α)^2^ ) [31]. The Sobel test was suitable as an alternative to bootstrapping because of the relatively large sample size [31,32]. All mediators were added separately, resulting in 12 single mediation models. For the significant mediators, the percentage mediating the association between family SES and children’s screen-time was calculated by dividing αβ by the c’-coefficient. The dataset generated and analyzed during the current study is available as Appendix A.

## 3. Results

### 3.1. Study Characteristics

Table 2 provides an overview of sample characteristics and descriptive statistics of variables for lower and higher SES families separately.

### 3.2. Association between Family SES and Children’s Screen-Time (τ-Coefficient)

A significant negative association was found between family SES and children’s screen-time (τ = −0.34, SE = 0.14, 95% CI = −0.62; −0.06). Children from lower SES families had higher levels of screen-time (lower SES: 2.41 ± 1.16 h/day; higher SES: 2.01 ± 0.96 h/day).

### 3.3. Associations between Family SES and Potential Mediators (α-Coefficients)

Family SES was only associated with the following three potential mediators: (a) being consistent concerning gaming, (b) giving an explanation about rules for gaming, and (c) avoiding negative role modeling concerning TV-time (*p* < 0.05; Table 3). Parents from higher SES families were less consistent concerning rules on gaming, gave an explanation about rules for gaming less often, and avoided more negative role modeling concerning TV-time.

### 3.4. Associations between Potential Mediators and Children’s Screen-Time (β-Coefficients)

The conceptual theory tests revealed significant negative associations between all potential mediators and children’s screen-time (*p* < 0.05; Table 3). When parents adopted more positive parenting practices, children had a lower screen-time: having rules about TV-time and gaming, following up these rules, giving an explanation about these rules, monitoring TV-time and gaming, motivating the child to watch less TV and to play less games, and avoiding negative role modeling concerning TV-time and gaming were related to a lower screen-time of their child.

### 3.5. Mediating Effect of Parenting Practices on the Relation between Family SES and Children’s Screen-Time (αβ-Coefficients)

Being consistent concerning rules about gaming and avoiding negative role modeling concerning TV-time significantly mediated the association between family SES and children’s screen-time (Table 3). Parents from lower SES families were more consistent concerning rules about gaming, which was associated with a higher screen-time in their children (αβ = 0.127; SE = 0.055). Parents from higher SES families avoided more negative role modeling, which was associated with a lower screen-time in their children (αβ = −0.082; SE = 0.040). The proportion mediated by being consistent concerning rules about gaming and avoiding negative role modeling concerning TV-time was −54.6% (suppressing effect) and 22.8%, respectively.

## 4. Discussion

To our knowledge, the present study is the first to explore the mediating effect of specific parenting practices on the association between family SES and young primary schoolchildren’s screen-time in a specific subgroup of families that are at an increased risk of developing type 2 diabetes. First, this study confirmed previous findings from studies in the general population that children from higher SES families engage in lower levels of screen-time [2,7,8,9,10]. Therefore, even in a specific, vulnerable study population, family SES plays an indispensable role. In addition, our results showed that children had lower levels of screen-time when parents had positive parenting practices. The evidence for a relationship between parenting practices and children’s screen-time is currently weak or mixed in the general population [15]. This important association between specific parenting practices and children’s screen-time could to be taken into account when designing future interventions for both lower and higher SES families identified to be at high risk for developing type 2 diabetes. Parents might be thought to apply rules about TV-time and gaming, to be consistent about those rules and to give an explanation about them, to monitor the child’s behavior (TV-time and gaming), to motivate the child to limit TV-time and gaming, and to avoid being a negative role model concerning both TV-time and gaming in order to reduce their primary schoolchild’s screen-time. 

When looking at the associations between family SES and parenting practices, family SES was only associated with three parenting practices, of which merely two were significant mediators in the association between family SES and children’s screen-time. First, parents of higher SES families avoided negative role modeling regarding TV-time more than parents of lower SES families. This might suggest that they watch less TV nearby their child, as was also found in the review of Gebremariam et al. [10]. Furthermore, when parents from higher SES families avoided negative role modeling regarding TV-time more, their children had a lower screen-time. In other studies, parent’s TV-time was also found to mediate the relationship between parental education and children’s/adolescents’ screen-time [15,33]. Therefore, in families with a high risk of developing type 2 diabetes, parents from lower SES families can be made aware of their function as role models towards their child. Parents can limit their own TV-time nearby their child to improve their child’s screen-time and help to reduce socio-economic differences in screen-time. 

Next, unexpectedly, parents from lower SES families were more consistent concerning rules about gaming and gave an explanation about rules on gaming more often. Furthermore, being consistent concerning rules about gaming was the other significant mediator in the relation between family SES and children’s screen-time. Parents from lower SES families were more consistent concerning rules about gaming, which was associated with higher levels of screen-time in their children. A first possible explanation for this result is that the prevalence of gaming is rather limited in younger primary schoolchildren (i.e., 6- to 9-year-olds) [34] and potentially even more limited in younger primary schoolchildren of higher SES families. As those children might not engage a lot in gaming, it could be that parents are not always consistent about their rules concerning gaming. To put it differently, those parents might think it is not really necessary to be consistent about their rules because the behavior does not occur much. In addition, we have no qualitative data on which rules parents have regarding gaming. It could be, for example, that parents from higher SES families are less consistent about their rule to play games for half an hour per day, and parents from lower SES families more consistent about their rule to play games for four hours per day. This shortcoming related to using quantitative questionnaire items should be taken into account by future studies and the association between family SES and (being consistent about) rules and regulations on screen-time needs to be further explored before drawing conclusions. Another possible explanation is that there might be a discrepancy between the investigated variables in this study, as the specific parenting practices were related to TV-time or gaming, whereas the outcome was screen-time in general. It has indeed been shown that TV-time contributes more to children’s total screen-time compared to gaming [35]. However, screen-time (and not TV-time) was chosen as an outcome measure since the health guidelines are formulated for screen-time in general (i.e., the combination of TV and computer activities [6]). In addition to this, it would have been relevant to investigate parenting practices related to computer use (and not only TV-time and gaming); however, because of the content of the Movie Models intervention, this was not assessed. 

So, besides these two mediators, no other parenting practices mediated the relationship between family SES and young primary schoolchildren’s screen-time. Because this study only investigated parenting practices as possible mediators, future research could use the socio-ecological framework of Owen et al. [36] to identify other possible mediating factors. For example, other parental factors (e.g., parental social norm, parental attitudes towards screen-time), the home environment (e.g., the number of screens available at home and in children’s bedroom), or the perceived neighborhood environment (e.g., presence of and access to play facilities), can be investigated to understand the mechanism through which family SES influences children’s screen-time. 

A first strength of the current study is that specific parenting practices were examined, which gives more insight into parental correlates of children’s screen-time compared to more generally formulated parenting practices (e.g., limit setting, using praise and rewards). Another strength is the focus on a specific vulnerable population, namely families with a higher risk of developing type 2 diabetes, as (intervention) studies focusing on this group can eventually help to prevent or postpone the disease [37]. Nevertheless, the tight focus on this vulnerable group implies that our findings cannot be generalized to the general population. Additionally, most participants were mothers (78.0%), which also compromises generalizability. A second limitation of the study is that the mean values for the parenting practices were high, which shows the limited amount of variance within and between the SES groups. These high values may be the result of social desirability bias since the administered questionnaires were self-reported, which might have contributed to the lack of significant mediators as well. Furthermore, five-point Likert scales were used to measure specific parenting practices, which means that the absolute magnitudes of these variables cannot be provided. A final study limitation is the cross-sectional design, from which no causal conclusions can be drawn. This implies that our suggestions for future interventions are preliminary and should be confirmed by longitudinal or experimental research.

## 5. Conclusions

Against our expectations, the investigated specific parenting practices could hardly explain the association between family SES and children’s screen-time in our specific study population of families that had an increased risk of developing type 2 diabetes. Only avoiding negative modeling of TV-time seems to be an important parenting practice to target in future programs aimed at reducing SES disparities in these children’s screen-time. Therefore, future research should examine other possible mediating factors to understand the mechanism through which family SES influences children’s screen-time in this vulnerable population. 

## Figures and Tables

**Table 1 ijerph-15-02553-t001:** Formulations of the questionnaire items of the specific parenting practices related to screen-time.

Factor	Question Item
Rules regarding TV-time	In our family, there are rules for my child about watching TV or DVDs.
Rules regarding gaming	In our family, there are rules for my child about playing videogames, computer games, PlayStation, Nintendo, etc.
Being consistent concerning rules about TV-time	The rules for my child about watching TV or DVDs are followed up.
Being consistent concerning rules about gaming	The rules for my child about playing videogames, computer games, PlayStation, Nintendo, etc. are followed up.
Giving an explanation on the rules regarding TV-time	I explain to my child why there are rules about watching TV or DVDs.
Giving an explanation on the rules regarding gaming	I explain to my child why there are rules about playing videogames, computer games, PlayStation, Nintendo, etc.
Monitoring children’s TV-time	I monitor the time my child watches TV or DVDs.
Monitoring children’s gaming	I monitor the time my child plays videogames, computer games, PlayStation, Nintendo, etc.
Motivating children to reduce TV-time	I try to motivate my child to watch less TV or DVDs.
Motivating children to reduce gaming	I try to motivate my child to play less videogames, computer games, PlayStation, Nintendo, etc.
Avoiding negative role modeling regarding TV-time	I limit my own watching of TV or DVDs nearby my child.
Avoiding negative role modeling regarding gaming	I limit my own playing of videogames, computer games, PlayStation, Nintendo, etc. nearby my child.

**Table 2 ijerph-15-02553-t002:** Sample characteristics and descriptive statistics (*n* = 247) for lower and higher SES families separately.

	Lower SES Families (57.6%)		Higher SES Families (42.4%)	
	Mean ± SD	Percentage	Mean ± SD	Percentage
Sex of child (% girls)	-	47.1% girls	-	50.5% girls
Age of child (years)	8.32 ± 0.89	-	7.97 ± 0.85	-
Questionnaire completed by mother; father; stepfather; other	-	77.9; 20.0; 0.7; 1.4	-	77.5; 22.5; 0; 0
Age of completer of the questionnaire (years)	40.17 ± 6.66	-	39.47 ± 4.41	-
Screen-time of child (h/day)	2.41 ± 1.16	-	2.01 ± 0.96	-
	**Mean ± SD**	**Percentage Never; Mostly Not; Sometimes/Sometimes Not; Mostly; Always**	**Mean ± SD**	**Percentage Never; Mostly Not; Sometimes/Sometimes Not; Mostly; Always**
Rules TV ^a^	3.59 ± 1.09	2.9; 15.7; 23.6; 35.0; 22.9	3.46 ± 1.13	4.9; 18.6; 19.6; 39.2; 17.6
Rules gaming ^a^	3.91 ± 1.17	5.2; 9.0; 14.2; 32.8; 38.8	3.66 ± 1.20	6.9; 11.9; 16.8; 36.6; 27.7
Being consistent TV ^a^	3.79 ± 0.86	0; 8.9; 23.0; 48.1; 20.0	3.71 ± 0.77	0; 6.4; 28.7; 52.1; 12.8
Being consistent games ^a^	4.10 ± 0.85	0; 4.8; 16.9; 41.1; 37.1	3.78 ± 0.96	1.2; 9.6; 22.9; 42.2; 24.1
Giving explanation TV ^a^	3.88 ± 1.03	1.5; 8.9; 23.7; 31.9; 34.1	3.73 ± 0.98	1.1; 12.6; 21.1; 43.2; 22.1
Giving explanation gaming ^a^	4.14 ± 0.92	0; 4.8; 21.8; 28.2; 45.2	3.78 ± 1.04	2.3; 12.8; 15.1; 44.2; 25.6
Monitoring TV ^a^	3.15 ± 1.21	10.8; 22.3; 20.1; 34.5; 12.2	3.32 ± 1.10	7.8; 17.5; 17.5; 49.5; 7.8
Monitoring gaming ^a^	3.50 ± 1.25	10.2; 12.4; 16.1; 39.4; 21.9	3.58 ± 1.16	8.9; 9.9; 12.9; 50.5; 17.8
Motivating TV ^a^	3.68 ± 0.86	0.8; 8.4; 28.2; 47.3; 15.3	3.64 ± 0.91	2.0; 8.1; 29.3; 45.5; 15.2
Motivating gaming ^a^	3.73 ± 1.05	5.0; 6.7; 22.5; 42.5; 23.3	3.70 ± 1.02	4.3; 7.6; 22.8; 44.6; 20.7
Avoiding negative role modeling TV ^a^	3.19 ± 1.17	9.4; 18.8; 28.3; 30.4; 13.0	3.59 ± 1.24	5.9; 17.6; 16.7; 31.4; 28.4
Avoiding negative role modeling gaming ^a^	3.92 ± 1.16	9.8; 8.3; 7.6; 28.8; 45.5	4.14 ± 1.19	6.1; 6.1; 9.2; 24.5; 54.1

SD = standard deviation, h = hours, ^a^ five-point-scale with a higher value representing a higher level of parenting practice.

**Table 3 ijerph-15-02553-t003:** Mediating role of parenting practices on the association between family SES and children’s screen-time.

Single Mediation Models	Action Theory Tests ^a^		Conceptual Theory Tests ^b^		Mediating Effects		Proportion Mediated
	α (SE)	95 % CI for α	β (SE)	95 % CI for β	αβ (SE)	95 % CI for αβ	%
Rules TV	−0.186 (0.146)	−0.473, 0.101	**−0.278 (0.061)**	**−0.398, −0.157**	0.052 (0.042)	−0.031, 0.134	/
Rules gaming	−*0.286* (*0.157*)	−*0.596, 0.025*	**−0.176 (0.060)**	**−0.294, −0.058**	0.050 (0.033)	−0.013, 0.114	/
Being consistent TV	−0.162 (0.111)	−0.380, 0.056	**−0.408 (0.086)**	**−0.577, −0.238**	0.066 (0.047)	−0.027, 0.159	/
Being consistent gaming	**−0.356 (0.127)**	**−0.606, −0.105**	**−0.357 (0.086)**	**−0.527, −0.186**	**0.127 (0.055)**	**0.020, 0.234**	**−54.6**
Giving explanation TV	−0.137 (0.140)	−0.414, 0.139	**−0.185 (0.070)**	**−0.323, −0.047**	0.025 (0.028)	−0.029, 0.079	/
Giving explanation gaming	**−0.318 (0.139)**	**−0.593, −0.043**	**−0.200 (0.079)**	**−0.355, −0.045**	0.064 (0.037)	−0.010, 0.137	/
Monitoring TV	0.134 (0.154)	−0.169, 0.438	**−0.220 (0.058)**	**−0.335, −0.105**	−0.029 (0.035)	−0.098, 0.039	/
Monitoring gaming	0.026 (0.159)	−0.287, 0.340	**−0.134 (0.059)**	**−0.250, −0.018**	−0.003 (0.021)	−0.045, 0.038	/
Motivating TV	−0.027 (0.118)	−0.260, 0.206	**−0.315 (0.080)**	**−0.472, −0.158**	0.009 (0.037)	−0.064, 0.081	/
Motivating gaming	−0.019 (0.144)	−0.303, 0.265	**−0.159 (0.073)**	**−0.303, −0.015**	0.003 (0.023)	−0.042, 0.048	/
Avoiding negative role modeling TV	**0.376 (0.157)**	**0.066, 0.686**	**−0.218 (0.057)**	**−0.331, −0.105**	**−0.082 (0.040)**	**−0.161, −0.003**	**22.8**
Avoiding negative role modeling gaming square ^c^	1.811 (1.126)	−0.408, 4.031	**−***0.017* (*0.009*)	−*0.034*, *0.000324*	−0.031 (0.025)	−0.080, 0.018	/

^a^ Associations between family SES and parenting practices; ^b^ associations between parenting practices and children’s screen-time, adjusting for family SES; ^c^ the variable ‘avoiding negative role modeling concerning gaming’ was square transformed to obtain a normal distribution; SE = standard error, CI = confidence interval. The proportion mediated was only calculated if the mediation effect was significant at the 95% level. Significant associations are presented in bold font; borderline significant associations are presented in italic font. All analyses were adjusted for child’s age and child’s gender.

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
