# Peer review of "Parenting Practices as a Mediator in the Association Between Family Socio-Economic Status and Screen-Time in Primary Schoolchildren: A Feel4Diabetes Study"

_ijerph, 2018, doi:10.3390/ijerph15112553_

Reviewer 1 Report

 The study is focused on an important issue of parenting practices and children screen-time. the manuscript is well written, with very minor issues (sometimes using synonyms instead of "current" and "currently" / "developing" and "developing" in the same sentence, such as in lines 232-235, may improve readability).

While the paper has a logic flow and provides sufficient details in describing variables, the research seems to be non-experimental, so the reference to intervention group (lines 107-108) and random assignment (line 107) is confusing and needs clarification. In addition, Table 1 may provide more relevant information for readers if the descriptives are broken by the dependent variable (family SES).

Assuming that a reader is not an expert in product of coefficients approach, a bit more of the explanation for methods along with reporting model fit may help with a better understanding. For example, clarifying on the adjustments related to use of more than one mediator may be considered.

Author Response

Point-by-point response to reviewer 1:

1. The study is focused on an important issue of parenting practices and children screen-time. the manuscript is well written, with very minor issues (sometimes using synonyms instead of "current" and "currently" / "developing" and "developing" in the same sentence, such as in lines 232-235, may improve readability).

Answer: We sincerely thank the reviewer for the compliments on the paper and for the valuable comments which has enabled us to further improve our manuscript. Please find hereafter our response to the two comments. In addition, we have thoroughly gone through the manuscript to check for readability issues.

2. While the paper has a logic flow and provides sufficient details in describing variables, the research seems to be non-experimental, so the reference to intervention group (lines 107-108) and random assignment (line 107) is confusing and needs clarification. In addition, Table 1 may provide more relevant information for readers if the descriptives are broken by the dependent variable (family SES).

Answer: Because we used the baseline data from a larger experimental study, we decided to describe the recruitment procedure of that study. However, to clarify this for the reader, we have added following information to the Methods section:

“2.2 Participants and Recruitment

As we use the baseline data from the Movie Models intervention that was integrated within the F4D- intervention, we describe the recruitment procedure of the larger experimental study. However, it should be noted that parents from both the intervention and control group were included in our study sample without making any distinction between both groups. A standardized, multi-stage sampling approach was applied for the recruitment of families from the provinces East- and West-Flanders (Belgium). (…)  In November-December 2016, the ‘high-risk families’ from both the intervention and control group filled out a questionnaire on parenting practices for the baseline measure of the Movie Models intervention (response rate = 53.9% = 247/458).”

Further, as suggested by the reviewer, we have adjusted Table 2 by describing all relevant variables for lower and higher SES families separately (see Table 2).

3. Assuming that a reader is not an expert in product of coefficients approach, a bit more of the explanation for methods along with reporting model fit may help with a better understanding. For example, clarifying on the adjustments related to use of more than one mediator may be considered.

Answer: We have now added following information about the product of coefficients approach, and specified that we assessed single mediator models:

“The mediation analyses (presented in Figure 1) consisted of the following steps. Firstly, the main association or direct association between family SES and children’s screen-time was examined (τ-coefficient or c-path). In the second stage, the mediating role of specific parenting practices related to screen-time was examined using the product-of-coefficients test of MacKinnon et al. [30]. This test included the following steps: (1) estimation of the associations between family SES and potential mediators (Action Theory test; α- coefficients or a-path); (2) estimation of the associations between the potential mediators and children’s screen-time (Conceptual Theory Test; β-coefficients or b-path), adjusting for family SES; and (3) calculation of the product of-coefficients (αβ) or c’-path, representing the mediated effect. The rationale behind this third step is that the mediation depends on the extent to which the predictor is related to the mediators, and the extent to which the mediators affect the outcome. After conducting these three steps, statistical significance of the mediated effect was estimated by dividing αβ by its standard error (SE), from which the outcome conforms to a z-distribution. If the ratio was more than 1.96 or smaller than -1.96, the indirect effect was indeed significant, which implied mediation. To calculate SE, the Sobel test was used: SE (αβ) = √(α2 *SE (β) 2 + β2 *SE (α) 2 ) [31]. The Sobel test was suitable as an alternative of bootstrapping because of the relatively large sample size [31, 32]. All mediators were added separately, resulting in twelve single mediation models.”

Reviewer 2 Report

This study reports the results of a intervention study at a time point. The description in this paper was done well and the study was carried out well. There were good statistical analyses and reporting was appropriate. The results were to the point and the discussion was appropriate for a paper that can be read by other readers. 

Author Response

Response to reviewer 2:

This study reports the results of an intervention study at a time point. The description in this paper was done well and the study was carried out well. There were good statistical analyses and reporting was appropriate. The results were to the point and the discussion was appropriate for a paper that can be read by other readers. 

Answer: We sincerely thank the reviewer for the compliments on the paper.